# TANGO2 is an acyl-CoA binding protein

Agustin Leonardo Lujan[1], Ombretta Foresti[1], Jose Wojnacki[1], Gonzalo Bigliani[1], Nathalie Brouwers[1], Maria Jesus Pena[1], Stefania Androulaki[1], Tomomi Hashidate-Yoshida[2], Maria Kalyukina[3], Sergey S. Novoselov[4], Hideo Shindou[2,5], and Vivek Malhotra[1,6,7]

**Loss of TANGO2 in humans precipitates metabolic crises during periods of heightened energy demand, such as fasting, infections, or high fever. TANGO2 has been implicated in various functions, including lipid metabolism and heme transport, and its cellular localization remains uncertain. In our study, we demonstrate that TANGO2 localizes to the mitochondrial lumen via a structural region containing LIL residues. Mutations in these LIL residues cause TANGO2 to relocate to the periphery of lipid droplets. We further show that purified TANGO2 binds acyl-coenzyme A, and mutations in the highly conserved NRDE sequence of TANGO2 inhibit this binding. Collectively, our findings suggest that TANGO2 serves as an acyl-coenzyme A binding protein. These insights may provide new avenues for addressing the severe cardiomyopathies and rhabdomyolysis associated with defective TANGO2 in humans.**

## Introduction

TANGO genes were first identified in 2006, with TANGO1 extensively studied for its role in collagen secretion (Bard et al., 2006; Saito et al., 2009; Saxena et al., 2024). TANGO2, another gene in this group, has gained significant attention due to its association with neurodevelopmental delay, hypothyroidism, rhabdomyolysis, and cardiomyopathy with life-threatening arrhythmias (Miyake et al., 2018). Patients with TANGO2 mutations often experience metabolic crises characterized by hypoglycemia, hyperlactatemia, and elevated long-chain acylcarnitines (Powell et al., 2021; Yokoi et al., 2022; Restrepo-Vera et al., 2023). These crises are notably triggered by energy stress conditions such as fasting, high fever, or infections (Miyake et al., 2022). A research foundation has been established to address this critical issue (https://www.tango2research.org). We have previously shown that cells lacking TANGO2 exhibit increased total lipid content, larger lipid droplets (LDs), and altered lipid catabolism (Lujan et al., 2023). Recent studies have also reported an imbalance in lipid profiles in other experimental models of TANGO2 (Kim et al., 2023; Mehranfar et al., 2024). However, the precise role of TANGO2 in lipid metabolism and its connection to the associated human pathologies remain unclear.

It is known that under conditions of nutrient availability, long-chain fatty acids (LCFAs) are stored in LDs and membranes and are utilized for energy production during fasting when glucose is limited (Bosch et al., 2020). LCFAs are transferred from LDs to mitochondria for energy production through a process called fatty acid β-oxidation (FAO) (Rambold et al., 2015). Even in conditions of sufficient glucose availability, FAO remains the primary energy source for tissues with high energy demands, such as the heart, muscles, and kidneys (Houten et al., 2016). Mitochondria receive lipids through nonvesicular exchanges to facilitate FAO (Lees and Reinisch, 2020). This transfer is mediated by lipid transfer proteins (Chiapparino et al., 2016). Long-chain acyl-coenzyme A (LC-CoA) thioesters, which are the active form of LCFAs, are highly dynamic in response to metabolic requirements. Stressful conditions like fasting or hypoxia result in a significant increase in LC-CoA levels in the liver (Yang et al., 2019). Perturbations in LCFA metabolism induce an increase in free acyl molecules with proinflammatory and lipotoxic properties that alter the activity of receptors, transporters, and enzymes (McCoin et al., 2015). These findings highlight the importance of LCFAs and transfer proteins in mitochondrial lipid metabolism and physiology.

We now demonstrate that TANGO2 binds acyl-CoA and elucidate the mechanism of its localization to the mitochondrial lumen. TANGO2 thus emerges as a potential new shuttle for intracellular trafficking acyl-CoA. In the absence of TANGO2, the mitochondrial acyl-CoA pool required for lipid metabolism, especially in conditions of nutrient starvation, is disrupted, resulting in impaired mitochondrial function. This function of TANGO2 as a carrier of acyl-CoA could help understand pathologies linked to dysfunctional TANGO2.

[1]Centre for Genomic Regulation (CRG), The Barcelona Institute for Science and Technology, Barcelona, Spain; [2]Department of Lipid Life Science, National Center for Global Health and Medicine (NCGM), Shinjuku-ku, Japan; [3]Department of Clinical and Experimental Epilepsy, UCL Queen Square Institute of Neurology, London, UK; [4]Department of Neuromuscular Diseases, UCL Queen Square Institute of Neurology, London, UK; [5]Department of Medical Lipid Science, Graduate School of Medicine, The University of Tokyo, Bunkyo-ku, Japan; [6]Universitat Pompeu Fabra (UPF), Barcelona, Spain; [7]ICREA, Barcelona, Spain.

Correspondence to Vivek Malhotra: vivek.malhotra@crg.eu.

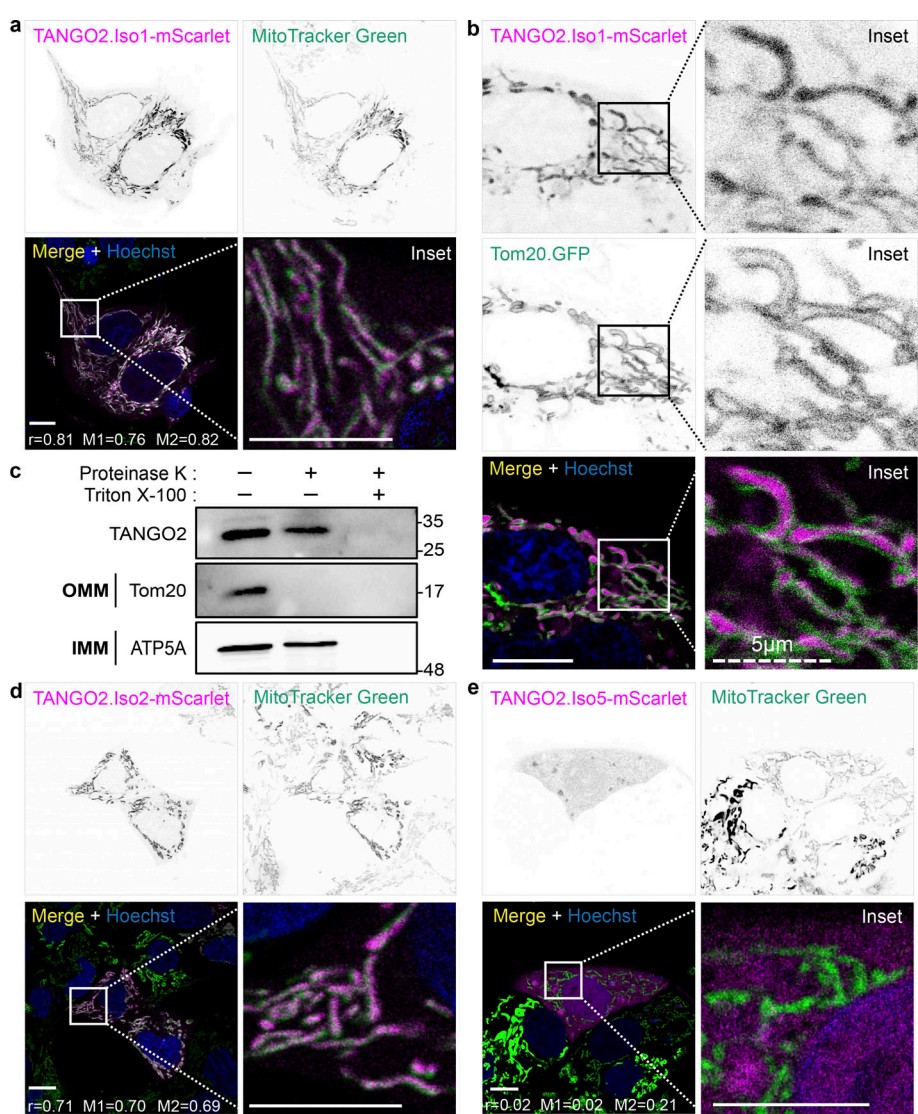

Figure 1. **Localization of TANGO2 isoforms 1, 2, and 5. (a)** HepG2 cells expressing TANGO2.Iso1-mScarlet (magenta) were incubated with Hoechst-33342 (blue) to detect DNA and MitoTracker Green to visualize the mitochondria. Pearson's correlation (r) and Mander's overlap (M1 and M2) coefficients were calculated with the coloc2 plugin in ImageJ software. **(b)** Cells cotransfected with TANGO2.Iso1-mScarlet (magenta) and Tom20.GFP (green) were incubated with Hoechst-33342 (blue) to detect DNA. **(c)** HepG2 cells were processed to obtain mitochondria-enriched fractions. The mitochondrial fractions were treated (+) with 50 µg/ml proteinase K for outer membrane protein cleavage and 1% Triton X-100 for total membrane disruption. Samples were resolved by SDS-PAGE and incubated with the antibodies Tom20, sited in the outer mitochondrial membrane, and ATP5A, sited in the inner mitochondrial membrane, and TANGO2. OMM, outer mitochondrial membrane; IMM, inner mitochondrial membrane. **(d and e)** HepG2 cells expressing TANGO2.Iso2-mScarlet (d) or TANGO2.Iso5-mScarlet (e) were incubated with MitoTracker Green to detect the mitochondria and Hoechst-33342 (blue) to label DNA. Pearson's correlation (r) and Mander's overlap (M1 and M2) coefficients were calculated with the coloc2 plugin in ImageJ software. Squares indicate the magnification area (inset). Scale bars = 10 µm. The white color indicates colocalization between magenta and green channels. Images are representative of three independent experiments. Source data are available for this figure: SourceData F1.

## Results

### TANGO2 is localized within the mitochondrial lumen

We previously showed that a pool of TANGO2 colocalizes with the mitochondrial marker MitoTracker by live-cell confocal microscopy and with the mitochondrial membrane protein Tom20 by Forster resonance energy transfer microscopy in fixed cells. To further ascertain the location of TANGO2 at mitochondria, we visualized HepG2 cells expressing tagged TANGO2 isoform 1-mScarlet (TANGO2.Iso1-mScarlet) using time-lapse confocal microscopy, followed by Pearson's correlation (r) and Mander's overlap (M1 and M2) coefficient analysis. Hoechst-33342 (blue) was used to detect DNA. TANGO2.Iso1-mScarlet fluorescence revealed high colocalization values with MitoTracker Green (r = 0.81; M1 = 0.76 and M2 = 0.82), thereby confirming our prior findings regarding the localization of TANGO2 (Fig. 1 a). Subsequently, we examined the location of TANGO2.Iso1-mScarlet in relation to the outer mitochondrial membrane protein Tom20. We generated a GFP-tagged transmembrane domain of the Tom20 plasmid to observe the outer membrane of mitochondria. Interestingly, the majority of TANGO2.Iso1-mScarlet was found within the region of

mitochondria enclosed by Tom20. These findings strongly suggest the presence of TANGO2.Iso1-mScarlet within the mitochondrial lumen (Fig. 1 b and Video 1).

To further validate the localization of TANGO2, we isolated a mitochondrial-enriched membrane fraction from HepG2 cells. This fraction was incubated with 50 µg/ml of proteinase K alone or proteinase K and 1% vol/vol Triton X-100. Following incubation for 1 h at 4°C, the proteinase activity was quenched by adding 2 mM phenylmethylsulfonyl fluoride (PMSF) for 20 min at 4°C. The samples were western-blotted with antibodies to TANGO2, the outer mitochondrial membrane protein Tom20, and the inner mitochondrial membrane protein ATP5A (Fig. 1 c). Our findings show that a significant pool of TANGO2 is protected from proteolysis in the absence of detergent. These collective data provide robust evidence supporting the localization of TANGO2 to mitochondria, with a substantial portion residing within the mitochondrial lumen.

### Mechanism of mitochondrial retention of TANGO2

There are at least six known isoforms of TANGO2 in humans, and isoform 1 is the most predominant and conserved isoform

across species. Isoforms 1 and 2 share high sequence similarity and are divergent from isoform 5. A sequence alignment of these three TANGO2 isoforms is shown in Fig. S1 a. We visualized the locations of these TANGO2 isoforms tagged with mScarlet in HepG2 cells. The specific constructs (magenta) were expressed by transient transfection, and HepG2 cells were then incubated with MitoTracker Green (green) for 30 min at 37°C to detect mitochondria followed by confocal live-cell imaging. While isoforms 1 and 2 were predominantly localized to mitochondria, isoform 5 was primarily cytosolic (Fig. 1, a, d, and e). Also, greater colocalization with mitochondria was observed with TANGO2 isoform 1 compared with TANGO2 isoform 2 (Fig. 1 d; r = 0.71; M1 = 0.70 and M2 = 0.69). No colocalization was observed between MitoTracker and TANGO2 isoform 5 (Fig. 1 e; r = 0.02; M1 = 0.02 and M2 = 0.21).

The primary amino acid sequence analysis revealed that isoforms 1 and 2 shared highly conserved N-terminal 60 residues, which are absent in isoform 5 (Fig. S1 a). We reasoned that this region might be responsible for targeting isoforms 1 and 2 to the mitochondria. To investigate further, we examined the conservation of these N-terminal residues across five TANGO2 orthologs. Our analysis revealed two distinct regions containing conserved LIL and NRDE sequences in the N-terminal 60 residues among the species evaluated (Fig. S1 b).

We generated mutants of TANGO2.Iso1-mScarlet using site-directed mutagenesis to replace NRDE with AGAA (TANGO2.ΔNRDE-mScarlet) and visualized the corresponding protein using time-lapse confocal microscopy. Cells were incubated with MitoTracker Deep Red (cyan) to detect mitochondria and Bodipy 493/503 (green) to mark LDs. The data show that mutant TANGO2.ΔNRDE-mScarlet (magenta) is localized to mitochondria like the wild-type TANGO2.Iso1-mScarlet (Fig. 2, a and b).

Subsequently, we changed the LIL residues to AGA in TANGO2.Iso1-mScarlet (TANGO2.ΔLIL-mScarlet) and visualized the cells with microscopy. The TANGO2.ΔLIL-mScarlet (magenta) mutant was almost entirely located in large punctae closely apposed to and, in many instances, partially merged with the rims of LDs (Fig. 2 c).

To corroborate whether TANGO2 mutants' localization might be cell-specific, we expressed the same mutant proteins into U2OS cells and observed the same pattern of its location as in the HepG2 cell line (Fig. S1, c–e).

The 40 amino-terminal residues of TANGO2 were recently found to play a role in localizing the protein to mitochondria in HeLa cells (Milev et al., 2021). To test that the LIL region determines the localization of TANGO2 to mitochondria, we generated a plasmid of the 40 N-terminal residues of TANGO2 tagged with the fluorescent protein mScarlet at the C terminus (TANGO2.40aa-mScarlet). We also generated a mutant by site-directed mutagenesis of LIL (LIL→AGA) in the background of TANGO2.40aa-mScarlet (TANGO2.40aa.ΔLIL-mScarlet). The wild-type tagged 40 N-terminal amino acid was localized to the mitochondria (Fig. 2 d; r = 0.79; M1 = 0.85 and M2 = 0.89), consistent with the findings of Milev et al. (2021). In contrast, the mutants in the LIL region revealed a cytosolic and nuclear distribution (Fig. 2 e; r = 0.31; M1 = 0.53 and M2 = 0.28).

## TANGO2 isoform 1 interacts with LDs

The change in the localization of TANGO2 after targeted mutagenesis of the LIL region prompted us to analyze its association with LDs. To confirm that the structures contacted by TANGO2 were LDs, we co-expressed full-length TANGO2.Iso1-mScarlet or mutant TANGO2, and ΔLIL-mScarlet constructs (magenta), with an LD marker construct (GPAT4.hairpin-NG; green) in HepG2 cells, followed by confocal live-cell imaging. The juxtaposition of wild-type TANGO2.Iso1 to LDs is dynamic in normal conditions (Fig. 2 f and Video 2; r = 0.06; M1 = 0.35 and M2 = 0.35). However, mutation of the LIL region changes the TANGO2 location and increases its interaction with LDs in the cytoplasm (Fig. 2 g and Video 3). We quantified the colocalization of TANGO2.ΔLIL-mScarlet and GPAT4.hairpin-NG fluorescence channels. Using coloc2 from ImageJ software, we calculated the Pearson correlation (r = 0.55) and Mander's overlap (M1 = 0.73 and M2 = 0.83) coefficients. Mander's score indicates that ~75% of selected TANGO2 and GPAT4 channels colocalize.

When TANGO2 is mutated, it also loses its mitochondrial location and tends to associate with LDs. This effect led us to consider a physiological situation where TANGO2 could change its location. As mentioned, fasting induces a two- to threefold increase in LC-CoA levels in rat liver. Similarly, after 20 min of ischemic perfusion in rat hearts, LC-CoA levels increase by 2–10-fold, and free CoA levels decrease (Yang et al., 2019). We investigated whether the location of TANGO2 is influenced by nutrient deprivation. Before depriving the cells of glucose, we treated the cells with 10 µg/ml cycloheximide for 1 h to inhibit new protein synthesis, maintaining this concentration throughout the experiment. Then, HepG2 cells were cultured in low-glucose media for varying periods (hour). The cells were then mechanically ruptured and fractionated. We evaluated the cytoplasmic pool and whole-cell lysate to compare TANGO2 protein content changes under fasting conditions. β-Actin was used as a loading control (Fig. 2 h). After 3 h of glucose starvation, the TANGO2 protein in the cytosolic pool increased up to threefold, consistent with the described modifications in LC-CoA levels (Fig. 2 i).

Our findings suggest that when TANGO2 is forced to lose its mitochondrial location, it preferentially associates with LDs in the cytoplasm. Regarding this, it is interesting to note that loss of TANGO2 results in growth of LDs as shown previously. One possibility is that TANGO2 extracts lipids for their transport to mitochondria.

## TANGO2 is an acyl-CoA binding protein

Our previous work demonstrated that TANGO2 has a role in lipid metabolism. Loss of TANGO2 increased LDs' size and affected phospholipid metabolism and fatty acid consumption. One fundamental change of note was the accumulation of lysophospholipids and the reduction in levels of phospholipids (Lujan et al., 2023). Thus, TANGO2 could function as (1) an enzyme with lysophospholipid acyltransferase (LPLAT) (Valentine et al., 2022) activity or (2) a lipid transfer protein.

To test the potential acyltransferase activity of TANGO2, we produced a C-terminal tagged version of TANGO2 isoform 1 by in-frame fusion with the Flag (DYKDDDDK) peptide

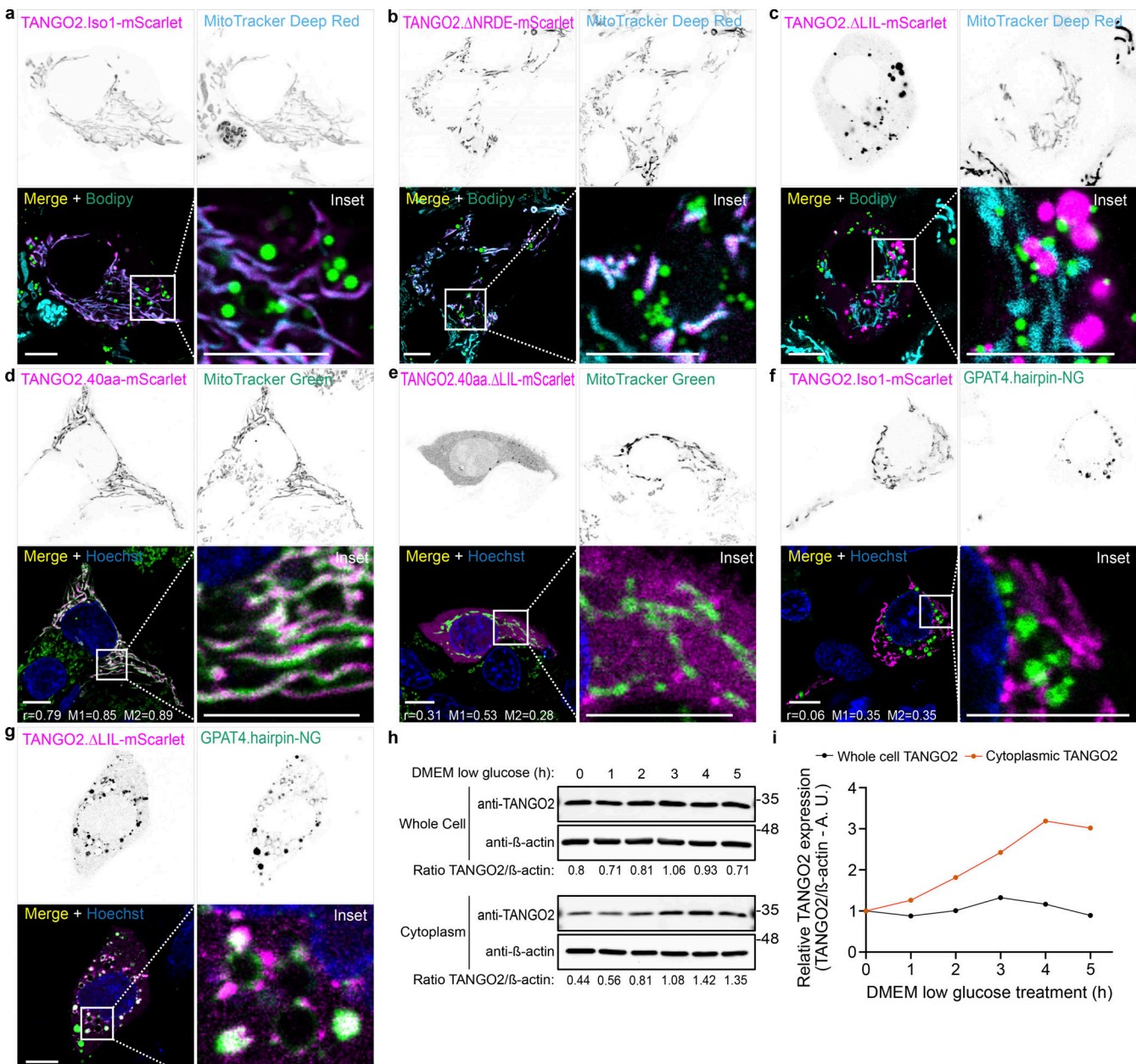

Figure 2. **TANGO2 wild-type and mutant location. (a–c)** HepG2 cells transfected with TANGO2.Iso1-mScarlet (a), or the mutants TANGO2.ΔNRDE-mScarlet (b) and TANGO2.ΔLIL-mScarlet (c) were incubated with the LD marker Bodipy Green and the mitochondrial marker MitoTracker Deep Red (cyan). **(d and e)** HepG2 cells expressing the mutant TANGO2.40aa-mScarlet (d) or TANGO2.40aa.ΔLIL-mScarlet (e) were incubated with MitoTracker Green to detect the mitochondria and Hoechst-33342 (blue) to visualize DNA. **(f and g)** HepG2 cells expressing TANGO2.Iso1-mScarlet (f) or TANGO2.ΔLIL-mScarlet (g) were co-expressed with the LD marker construct GPAT4.hairpin-NG (green) and incubated with Hoechst-33342 (blue) to visualize DNA. Pearson's correlation (r) and Mander's overlap (M1 and M2) coefficients were calculated with the coloc2 plugin in ImageJ software. Squares indicate the magnification area (inset). Scale bars = 10 μm. Images are representative of three independent experiments. **(h)** HepG2 cells were incubated in a low-glucose medium for different periods (hour) and mechanically lysed, and the cytoplasmic and whole-cell fractions were analyzed by western blot. β-Actin was used as a loading control. **(i)** Graph shows the relative expression of the TANGO2/β-actin ratio in the cytoplasmic and whole-cell fractions. Source data are available for this figure: SourceData F2.

(TANGO2.Iso1-Flag). Then, we transfected the TANGO2.Iso1-Flag construct or an empty vector in Chinese hamster ovary (CHO-K1) cells. After 48 h, we collected mitochondrial fraction and performed the LPLAT activity assay using C16:0 lyso-phosphatidate (LPA)/lysophosphatidylcholine (LPC)/lysophos-phatidylethanolamine (LPE)/lysophosphatidylserine (LPS)/lysophosphatidylglycerol (LPG)/lysophosphatidylinositol (LPI)

as an acyl acceptor, and C16:0-coenzyme A (CoA), C18:1-CoA, C18:2-CoA, C20:4-CoA, and C22:6-CoA as acyl donors. We determined the enzyme LPLAT enzymatic activity by measuring specific phospholipid production using a liquid chromatograph–mass spectrometer (LCMS). In the TANGO2-overexpressed samples (vermillion bars), no significant changes were observed in the production of phosphatidic acid (PA; Fig. 3 a), phosphatidylcholine

Figure 3. **TANGO2 interacts with specific acyl chains. (a–f)** LPLAT assays were performed in CHO-K1 cell lysates using C16:0-LPA, LPC, LPS, LPI, LPE, or LPG as acyl acceptors, and C16:0-CoA, C18:1-CoA, C18:2-CoA, C20:4-CoA, and C22:6-CoA as acyl donors. LPLAT activity was tested by measuring PA (a), PC (b), phosphatidylserine (PS; c), phosphatidylinositol (PI; d), phosphatidylethanolamine (PE; e), or phosphatidylglycerol (PG; f) molecule production in TANGO2-overexpressed (vermillion bars) compared with control cells (blue bars) and background noise (gray). The data are representative of three independent experiments with similar results. **(g and h)** Real-time NBD fluorescence intensity analysis of 2 µM 16-NBD-16:0-CoA or 18-NBD-18:1-CoA alone (0–300 s), incubated with 1 µM wild-type TANGO2 or mutant TANGO2.ΔNRDE protein (300–600 s), and with 0.8% Tween-20 addition (1,200–1,800 s) at 24°C. All the reactions were performed in 150 µl of reaction buffer (Tris-HCl 20 mM, pH 7.4; black). **(i and j)** Real-time NBD fluorescence intensity analysis of 1 µM 16-NBD-16:0-CoA (i, black) or 18-NBD-18:1-CoA (j, black) incubated with 1 µM 16:0-CoA (vermillion), 18:1-CoA (blue), or CoA alone (green) for 5 min (0–300 s), and then with 1 µM wild-type TANGO2 protein (330–630 s) at 24°C. The NBD fluorescence intensity experiments are representative of at least two independent experiments with similar results. Data are shown as the mean ± SD. A. U. means arbitrary units.

(PC; Fig. 3 b), phosphatidylserine (PS; Fig. 3 c), phosphatidylinositol (PI; Fig. 3 d), phosphatidylethanolamine (PE; Fig. 3 e), or phosphatidylglycerol (PG; Fig. 3 f) molecules compared with control cells (blue bars). TANGO2 is therefore not an acyltransferase protein under the experimental conditions tested.

To test whether TANGO2 binds lipids, we purified TANGO2.Iso1-Flag (TANGO2) and TANGO2.ΔNRDE.Flag (TANGO2.ΔNRDE) proteins from HEK293 cells as described in the Materials and methods section (Fig. S1, f and g). The purified proteins were tested for binding to lipids using a fluorescence-quenching assay. The basic tenet of this assay is that

fluorescence emission of N-[(7-nitro-2-1,3-benzoxadiazol-4-yl)-methyl]-amino-(NBD)-acyl-CoA is quenched upon specific binding to a protein (Soupene and Kuypers, 2015). We measured the fluorescence intensity of commercially available fluorescent acyl-CoAs 16-NBD-16:0-CoA or 18-NBD-18:1-CoA at 30-s intervals for three equal periods of 5 min, both in the absence and in the presence of the purified protein. After 5 min, we added 1 µM of the wild-type TANGO2.Flag or mutant TANGO2.ΔNRDE protein to quench the lipid fluorescence. Following this, we incorporated 0.8% Tween-20 to dissociate the binding of the protein to the lipid. Restoring fluorescence intensity to initial levels after the detergent addition reveals the disruption of the specific protein–lipid association. As shown in Fig. 3, the NBD fluorescence was significantly lower (P < 0.01) when 1 µM TANGO2 (vermillion) was added to 2 µM 16-NBD-16:0-CoA (Fig. 3 g) or 18-NBD-18:1-CoA (Fig. 3 h). However, experiments with the mutant TANGO2.ΔNRDE (blue) showed no interaction with the 16-NBD-16:0-CoA or 18-NBD-18:1-CoA.

To further investigate the binding specificity of TANGO2 to lipid chains, we repeated the assay described above, introducing competitors such as 16C-CoA and 18:1(n-12)-CoA, as well as CoA alone. We measured the fluorescence intensity of 16-NBD-16:0-CoA (Fig. 3 i) and 18-NBD-18:1-CoA (Fig. 3 j) at 30-s intervals for two periods of three min, both in the absence and in the presence of the competitors. We used each competitor at a final concentration of 1 µM. After 3 min, we added 1 µM of the wild-type TANGO2.Flag protein to quench the lipid fluorescence. The relative fluorescence intensity of 16-NBD-16:0-CoA was mainly reduced in the presence of 16C-CoA and to a lesser extent of 18:1(n12)-CoA (Fig. 3 i). For 18-NBD-18:1-CoA, the only competitor that decreased the interaction between TANGO2 and the NBD-acyl-CoA is 16C-CoA (Fig. 3 j). Remarkably, CoA alone did not displace the interaction between TANGO2 and acyl-CoA. These results demonstrate that TANGO2 interacts directly and specifically with the lipid portion of acyl-CoA molecules.

### TANGO2 forms oligomers and contains a molecular pocket that could accommodate palmitate

The interaction between TANGO2 and acyl-CoA encouraged us to examine its structural organization.

Purified TANGO2 was analyzed by mass photometry and revealed that TANGO2 assembles into oligomers (Fig. 4 a). To confirm this finding, we cotransfected TANGO2.Flag and TANGO2.mScarlet plasmids in HepG2 cells. After 48 h of expression, cells were collected and processed for immunoprecipitation as described in the Materials and methods section. We immunoprecipitated TANGO2.Flag using anti-flag affinity resin. As expected, TANGO2.mScarlet was found to co-immunoprecipitate with TANGO2.Flag, which suggests that TANGO2 forms a protein complex with itself (Fig. 4 b).

Lipid transfer proteins involved in nonvesicular transport are typically characterized by a globular and soluble structure (Chiapparino et al., 2016). We analyzed the predicted structure of TANGO2 isoform 1 (AF-Q6ICL3-F1) using AlphaFold, which generates a per-residue confidence score known as the predicted local distance difference test (pLDDT), ranging from 0 (low confidence) to 100 (maximum confidence) (Jumper et al., 2021).

For TANGO2 isoform 1, the pLDDT analysis revealed a very high confidence in the structural integrity of the residues (>90). The structure of TANGO2 isoform 1 is globular, featuring a core of two β-sheets surrounded by α-helices with a potential helix–turn–helix cap and a prominent cavity (Fig. 4 c and Video 4) similar to other lipid binding proteins (Yabut and Isoherranen, 2023; Chiapparino et al., 2016). We employed the molecular pocket predictor KVFinder (Guerra et al., 2023) to explore these cavities, which identified nine distinct cavities (designated A to I; violet) on the surface of TANGO2 (dark gray) (Fig. 4 d, Table S1, and Video 5). Further analysis of the TANGO2 architecture revealed that the NRDE region (orange) resides within the main molecular pocket (cavity F), while the LIL mitochondrial inter-actor domain (green) is located within the second largest cavity (cavity B) on the opposite face of the NRDE region (Fig. 4 e and Video 6). Finally, we used AlphaFold 3.0 to simulate the inter-action between TANGO2 and palmitic acid, a 16-carbon chain fatty acid. The simulation suggested that up to two palmitate molecules could be accommodated within the central cavity of TANGO2, with the hydrophilic head positioned near the NRDE domain and the hydrophobic tail fitting snugly inside the molecular pocket (Fig. 4 e and Video 7).

Our results, both experimentally and through simulation, demonstrate that TANGO2 interacts with lipid chains via the NRDE domain situated within the molecular pocket.

## Discussion

There are reports suggesting that TANGO2 is exclusively cytosolic (Jennions et al., 2019), with two studies proposing that it functions as a heme transporter (Sun et al., 2022; Han et al., 2023). However, the role of TANGO2 as a heme transporter has been called into question, leaving uncertainties about its precise localization and function (Sandkuhler et al., 2023a, Preprint).

### TANGO2 shuttles between the cytoplasm and the mitochondrial lumen

Our data demonstrate that TANGO2 is localized to the mito-chondria, with a significant portion residing within the mito-chondrial lumen. This mitochondrial localization depends on an evolutionarily conserved LIL sequence, as mutants lacking these amino acids are instead found in the cytoplasm, often closely associated with, or even merging with, LDs. Moreover, we observed an increase in the cytoplasmic pool of TANGO2 under glucose starvation. These findings clarify the mechanisms governing TANGO2's localization to mitochondria and the conditions that influence its distribution between mitochondria and the cytoplasm.

### TANGO2 is an acyl-CoA protein

TANGO2 deletion showed similar phenotypes across diverse cell lines, marked by the accumulation of neutral lipids and lyso-phospholipids over phospholipids (Lujan et al., 2023; Kim et al., 2023; Mehranfar et al., 2024). Acyltransferases are responsible for converting all lysophospholipids into their corresponding phospholipids through acylation. We have ruled out the

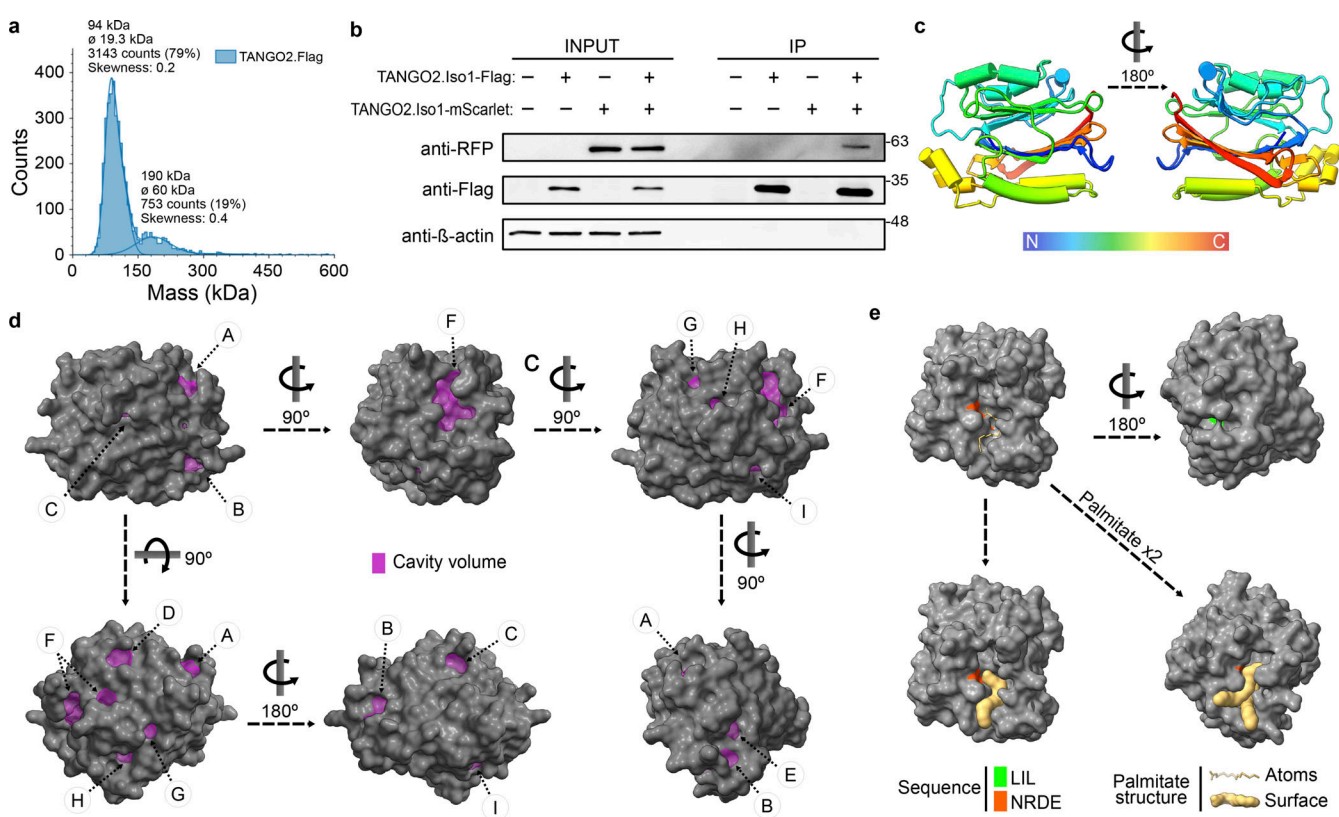

**Figure 4. Structure of TANGO2 and interaction simulation. (a)** Mass distribution for TANGO2.Flag oligomers by mass photometry. **(b)** HepG2 cells co-transfected with TANGO2.Flag and TANGO2.mScarlet were immunoprecipitated (IP) with anti-flag resin followed by western blot analysis. The left and right panels show cell lysate input and IP, respectively. The top membrane shows immunoblotting against RFP. Flag and β-actin were used as IP and loading controls. **(a–c)** Cylinder (α-helices) and arrow (β-sheets) cartoon representation of the TANGO2 isoform 1 structure. The rainbow-colored chain indicates the beginning of the amino-terminal sequence (blue) to the carboxyl-terminal sequence (red) of TANGO2. The N° indicates the rotation degrees of the protein structure in the y axis. **(d)** Visualization of the cavitation (violet) from KVFinder in the predicted three-dimensional surface of the TANGO2 isoform 1 protein. **(e)** Position of the LIL and NRDE sequences in the TANGO2 structure and interaction simulation with one and two palmitate chains by AlphaFold 3.0. All structural representations of TANGO2 are based on AlphaFold prediction and visualized using UCSF ChimeraX. Source data are available for this figure: SourceData F4.

possibility that TANGO2 is an enzyme with acyltransferase activity. However, the purified wild-type TANGO2 protein was found to specifically bind to the lipid portions of acyl-CoAs, particularly C16:0-CoA and C18:1-CoA, with a preference for the shorter fatty acid chain. In contrast, the mutant TANGO2.ΔNRDE did not show any direct interaction with these acyl-CoAs, highlighting the crucial role of the NRDE domain in the binding process.

The AlphaFold-predicted structure of TANGO2 reveals two key features. First, there is a distinct pocket within the TANGO2 structure, with the NRDE domain positioned inside it, capable of accommodating acyl molecules. Second, the LIL domain, essential for mitochondrial localization, is situated on the opposite face of the acyl-CoA binding pocket.

## TANGO2 deficiency and CoA metabolism deficiency

Inborn errors in mitochondrial acyl-CoA metabolism share clinical features with TANGO2 patients, including systemic decompensations, cardiomyopathy, rhabdomyolysis, cardiac arrhythmias, and the accumulation of cytoplasmic lipid–filled vacuoles. These conditions also exhibit similar biochemical profiles, such as hypoglycemia, hyperammonemia, hyperlactatemia,

and ketosis. Notably, the most severe episodes often occur after stressors such as fasting, exercise, or infections (Yang et al., 2019).

The carnitine shuttle has been suggested as the primary mechanism for transferring lipids into mitochondria. However, the intermediate components involved in the interaction of neutral lipids with carnitine palmitoyltransferases (CPT), the transport of lipids from CPT-I to CPT-II, and the transfer of individual phospholipids between mitochondria and other organelles remain poorly understood. It has been reported that vitamin B supplementation protects TANGO2 patients from experiencing metabolic crises (Sandkuhler et al., 2023b; Asadi et al., 2023; Miyake et al., 2023). Vitamin B5 (pantothenic acid) serves as the biological precursor for synthesizing CoA, which is essential for fatty acid metabolism in mitochondria. Humans obtain vitamin B5 through their diet, and its primary function is to produce CoA, which is then converted into acyl-CoA thioesters, key molecules in energy metabolism (Dibble et al., 2022; Goldford et al., 2017). It is possible that vitamin B5 supplementation increases the production of acyl-CoA, allowing even a small pool of acyl-CoA to be transported into the mitochondria in TANGO2-deficient patients, thereby supporting the residual metabolism of essential lipids. However, the underlying

mechanism behind the temporary relief provided by vitamin B5 remains unclear.

In summary, our data strongly suggest that TANGO2 is located within the mitochondrial lumen, and it binds acyl-CoA. This positions TANGO2 as a protein involved in acyl-CoA metabolism. However, it remains unclear whether TANGO2 binds acyl-CoA in the cytoplasm and transports it to the mitochondrial lumen to support the β-oxidation pathway, facilitates the presentation of acyl-CoA to specific enzymes involved in lipid metabolism, traffics to the mitochondrial lumen via the carnitine shuttle pathway, or utilizes a novel, stress-induced pathway.

Now that we know TANGO2 is an acyl-CoA binding protein, this discovery should help explain why the loss of TANGO2 function leads to severe conditions like cardiomyopathy and rhabdomyolysis.

# Materials and methods

## Cell culture
HepG2 and U2OS cells were cultured in Dulbecco's modified Eagle's medium (DMEM; Cat# SH30243.01; Lonza) cell culture media supplemented with 10% (vol/vol) heat-inactivated fetal bovine serum (FBS; Cat# 10270-106; Gibco) (DMEM complete), 100 U/ml penicillin, and 100 µg/ml streptomycin (Cat# 15140122; Gibco) (DMEM complete), at 37°C in a humidified incubator supplied with 5% $CO_2$. For glucose-starved conditions, cells were incubated in DMEM low-glucose, pyruvate cell culture media (Cat# 11885084; Gibco) without FBS for the indicated time (hour).

## Reagents
Primary antibodies used were as follows: anti-TANGO2 (Cat# NBP1-70463; Novus Biologicals), anti-Tom20 (Cat# sc-17764; Santa Cruz), anti-ATP5A (Cat# ab14748; Abcam), anti-β-actin (Cat# A1978; Sigma-Aldrich), and anti-Flag (Cat# F1804; Sigma-Aldrich). Secondary antibodies used were as follows: donkey anti-mouse IgG-HRP (Cat# 715035151; Jackson Immuno-Research), donkey anti-rabbit IgG-HRP (Cat# 715035152; Jackson ImmunoResearch), Alexa Fluor 680 donkey anti-rabbit (Cat# A10043; Invitrogen), and Alexa Fluor 800 donkey anti-mouse (Cat# A32789; Invitrogen). Reagents used were as follows: Hoechst-33342 (Cat# B2261; Sigma-Aldrich), MitoTracker Green (Cat# M7514; Thermo Fisher Scientific), MitoTracker Deep Red (Cat# M22426; Thermo Fisher Scientific), Bodipy Green (Cat# D3922; Invitrogen), proteinase K solution (Cat# AM2546; Ambion), Triton X-100 (Cat# T9284; Sigma-Aldrich), Tween-20 (Cat# P1379; Sigma-Aldrich), PMSF (Cat# P7626; Sigma-Aldrich), 16-0-LPA (Cat# 857123P; Avanti Polar Lipids), 16-0-LPC (Cat# 855675P; Avanti Polar Lipids), 16-0-LPE (Cat# 856705P; Avanti Polar Lipids), 16-0-LPS (Cat# 858142P; Avanti Polar Lipids), 16-0-LPG (Cat# 850102P; Avanti Polar Lipids), 16-0-LPI (Cat# 858122P; Avanti Polar Lipids), C16:0-CoA (Cat# 870716P; Avanti Polar Lipids), C18:1-CoA (Cat# 870719P; Avanti Polar Lipids), C18:2-CoA (Cat# 870736P; Avanti Polar Lipids), C20:4-CoA (Cat# 870721P; Avanti Polar Lipids), C22:6-CoA (Cat# 870728P; Avanti Polar Lipids), PA14:0/14:0 (Cat# 830845P; Avanti Polar Lipids), PC 14:0/14:0 (Cat# 850345P; Avanti Polar

Lipids), PE 14:0/14:0 (Cat# 850745P; Avanti Polar Lipids), PS 14:0/14:0 (Cat# 840033P; Avanti Polar Lipids), PI 18:0/18:0 (Cat# 850143P; Avanti Polar Lipids), PG 14:0/14:0 (Cat# 840445P; Avanti Polar Lipids), 16-NBD-16:0-CoA (Cat# 810705P; Avanti Polar Lipids), 18-NBD-18:1-CoA (Cat# 810229P; Avanti Polar Lipids), CoA (Cat# 870700P; Avanti Polar Lipids), Lipofectamine 2000 reagent (Cat# 11668019; Thermo Fisher Scientific), Lipofectamine 3000 reagent (Cat# L3000015; Thermo Fisher Scientific), proteinase inhibitor mixture cOmplete (Cat# 11836153001; Roche), fatty acid–free bovine serum albumin (BSA; Cat# A7030; Sigma-Aldrich), PMSF (Cat# P7626; Sigma-Aldrich), leupeptin hemisulfate (Cat# 101346; Focus Biomolecules), aprotinin (Cat# ab146286; Abcam), and pepstatin A (Cat# A2205; Panreac Química).

## Plasmids and cell transfection
For transient protein overexpression, cells were plated and 24 h later transfected with a 1:1.5 ratio (DNA plasmid:Lipofectamine) using Lipofectamine 3000 reagent following the manufacturer's recommendations. Plasmids generated in our laboratory were as follows: TANGO2.Iso1-mScarlet, TANGO2.Iso2-mScarlet, TANGO2.Iso5-mScarlet, TANGO2.ΔLIL-mScarlet, TANGO2.ΔNRDE-mScarlet, TANGO2.40aa-mScarlet, TANGO2.40aa.ΔLIL-mScarlet, TANGO2.Iso1-Flag, TANGO2.ΔHXXXXD-Flag, TANGO2.ΔLIL-Flag, TANGO2.ΔNRDE-Flag, ssTOMM20.GFP (Tom20.GFP), GPA-T4.hairpin-NeonGreen. 2 days after expression, cells were ready for real-time confocal microscopy assays.

## Live-cell imaging
HepG2 cells ($2.5 \times 10^5$ cells per dish) or U2OS ($1 \times 10^5$ cells per dish) were grown on 35-mm dishes with a polymer coverslip bottom for high-end microscopy (Cat# 81156; ibidi GmbH). After plasmid transfection, cells were washed, and the cell medium was replaced with fresh, complete, or conditioned medium for the indicated time. The cells were washed several times and incubated with reagents (Hoechst-33342, Bodipy Green, MitoTracker Green, MitoTracker Deep Red) in the appropriate medium for 30 min at 37°C in a 5% $CO_2$ atmosphere. Finally, after several washes, cells were used to perform the experiments described in the paper. High-resolution and time-lapse images were acquired in an inverted Leica STELLARIS confocal microscope equipped with photomultipliers and hybrid detectors. Samples were analyzed at 37°C in a 5% $CO_2$ atmosphere. Images were processed using ImageJ software. We calculated the Pearson correlation (r) and Mander's overlap coefficients (M1 and M2) of confocal images with the coloc2 plugin in ImageJ software for colocalization analysis.

## Protease protection assay in mitochondrial isolation
HepG2 cells ($1 \times 10^6$ cells per dish) were grown on a 10-cm dish with DMEM complete. After 48 h, cells were detached by trypsinization and collected at 800 $g$ for 5 min at 20°C. The cells were suspended in a homogenization buffer (250 mM sucrose, 20 mM Hepes, pH 7.4, 1 mM EDTA) and broke on ice using 22G and 27G needles (10 shots each). Cell homogenates were centrifuged at 300 $g$ for 5 min at 4°C to remove pellets containing unbroken cells and nuclei. The supernatant was centrifuged at 9,000 $g$ to

concentrate the crude mitochondrial pellet. The isolated mitochondria were incubated without or with 50 µg/ml proteinase K in the absence or presence of 1% Triton X-100 + 0.5% sodium deoxycholate for 1 h on ice. All the samples were incubated with 2 mM PMSF to stop the proteinase reaction. Finally, the samples were prepared for protein separation via SDS-PAGE.

## Immunoblotting

Immunoblot analysis was performed following the Bio-Rad general protocol recommendations. Briefly, equal amounts of protein were resolved by SDS-PAGE, transferred onto 0.45-µm PVDF membranes (Cat# 10600023; Amersham), and incubated overnight with anti-TANGO2 (1:1,000), anti-Flag (1:1,000), anti-Tom20 (1:1,000), anti-ATP5A (1:1,000), and anti-β-actin (1:10,000) primary antibodies followed by HRP- or Alexa-conjugated secondary antibodies (1:20,000) for 1 h at RT. Bands were visualized using Odyssey CLX (LI-COR Biosciences) or iBright CL1500 Imaging System (Thermo Fisher Scientific) equipment.

## Expression of TANGO2 in CHO-K1 cells

After 48 h of transfection with an empty vector or TANGO2.Flag cDNA in pcDNA3.1 using Lipofectamine 2000, cells from 10-cm dishes were scraped into 1 ml of ice-cold buffer containing 20 mM Tris-HCl (pH 7.4), 300 mM sucrose, and proteinase inhibitor mixture, and then sonicated three times on ice for 30 s. After centrifugation for 10 min at 800 $g$, each supernatant was collected and centrifuged at 9,000 $g$ for 10 min. The resulting pellets, which are mitochondrial fractions, were resuspended in a buffer containing 20 mM Tris-HCl (pH 7.4) and 1 mM EDTA. The protein concentration was measured by the method of Bradford, using a commercially prepared protein assay solution (Bio-Rad) and BSA (Fraction V, fatty acid–free; Sigma-Aldrich) as a standard.

## LPLAT assay

The LPAAT assay and the LPCAT/LPEAT/LPSAT/LPGAT/LPIAT assay as LPLAT assay were performed separately. For the LPAAT assay, proteins (1 µg) were incubated with 20 µM C16:0-LPA and each 20 µM acyl-CoA (C16:0-CoA, C18:1-CoA, C18:2-CoA, C20:4-CoA, and C22:6-CoA) in the presence of 100 mM Tris-HCl (pH 7.4), 1 mM EDTA, 2 mM CaCl$_2$, and 0.015% Tween-20 for 10 min at 37°C. For LPCAT/LPEAT/LPSAT/LPGAT/LPIAT assays, proteins (1 µg) were incubated with each 5 µM lysophospholipid (C16:0-LPC/LPE/LPS/LPG/LPI) and each 10 µM acyl-CoA (C16:0-CoA, C18:1-CoA, C18:2-CoA, C20:4-CoA, and C22:6-CoA) in the presence of 100 mM Tris-HCl (pH 7.4), 1 mM EDTA, 2 mM CaCl$_2$, and 0.015% Tween-20 for 10 min at 37°C. Then, the reaction was stopped by the addition of 300 µl of CHCl3:MeOH = 1:2 (vol/vol). Internal standard mix (50 µl of 1 µM PA14:0/14:0 for the LPAAT assay or 50 µl of 0.2 µM PC 14:0/14:0 and each 1 µM PE 14:0/14:0, DMPS 14:0/14:0, DSPI 18:0/18:0, DMPG 14:0/14:0) was added, and total lipids were extracted by the Bligh–Dyer method. Lipids were analyzed using LCMS-8050 (Shimadzu Corporation). All lipid reagents were purchased from Avanti Polar Lipids.

## TANGO2 protein purification

The TANGO2.Flag or TANGO2.ΔNRDE.Flag plasmid DNAs were used to transfect 30 ml cultures of Expi293 cells (Thermo Fisher Scientific) using ExpiFectamine 293 Transfection Kit (Cat# A14524; Gibco) following the manufacturer's instructions. Cells were cultured in Expi293 Expression Medium at 37°C with 8% CO$_2$ and 125 rpm platform shaking. After 20–30 h (depending on the efficiency of transfection), cells were pelleted at 1.7 rpm for 5 min, washed with PBS, flash-frozen in liquid nitrogen, and stored at –70°C. For purification, cell pellets were thawed and resuspended in 50 mM Hepes, pH 7.3, 175 mM NaCl, 1 mM EDTA, 1 mM PMSF, 1 mM TCEP, 0.5 µg/ml DNase, and a protease inhibitor cocktail (Roche). All subsequent steps were performed at room temperature. Cells were broken using a Dounce homogenizer followed by centrifugation (14,000 rpm for 30 min) to remove cellular debris. The supernatant was incubated with prewashed anti-Flag affinity resin (Cat# A2220; Sigma-Aldrich) for 1.5 h, followed by washes with the same buffer (excluding EDTA) and a 30-min ATP wash (1 mM ATP with 1 mM MgCl$_2$). The resin was then washed with 50 mM Hepes, pH 7.3, 175 mM NaCl, 1 mM EDTA, 5% glycerol, and 1 mM TCEP before the protein was eluted in fractions using 125 ng/ml of a 3xFlag peptide (Cat# A6001; APExBio). Eluted fractions were aliquoted, snap-frozen in liquid nitrogen, and stored at –70°C until needed.

## Protein–acyl-CoA binding and competition experiments

The in vitro analysis of TANGO2-acyl-CoA binding was accomplished in black 96-well optical-bottom plates (Cat# 165305; Thermo Fisher Scientific). In this assay were used the purified TANGO2.Flag or TANGO2.ΔNRDE.Flag and the fluorescently labeled lipids 16-NBD-16:0-CoA or 18-NBD-CoA 18:1. For direct binding, we used 2 µM of each NBD-acyl-CoA followed by measurement of the NBD (465/535 nm) fluorescence intensity. Then, a constant concentration of 1 µM protein was added and incubated at 24°C for 10 min in the dark, and the fluorescence intensity of the lipid was measured again. All reactions were performed in technical triplicate in 150 µl of 20 mM Tris-HCl, pH 7.4 (reaction buffer). A final concentration of 0.8% Tween-20 was added to disrupt specific protein–lipid binding. The analysis of fluorescence intensity was performed using the microplate reader Tecan Infinite M200 (Tecan) equipment.

For the competition assay, we used 1 µM of each NBD-acyl-CoA incubated with 1 µM 16:0-CoA, 18:1-CoA, or CoA alone, followed by measurement of the NBD (465/535 nm) fluorescence intensity. Then, we added 1 µM wild-type TANGO2 protein and incubated it at 24°C in the dark, and the fluorescence intensity of the lipid was measured again.

## Co-immunoprecipitation

HepG2 cells (1 × 10⁶ per dish) were grown on 60-mm dishes and transfected with TANGO2.Flag or TANGO2.mScarlet plasmids alone or together using Lipofectamine 3000 following the manufacturer's instructions. After 48 h, cells were washed twice with cold PBS, and 0.6 ml lysis buffer (50 mM Tris-HCl, 50 mM NaCl, 1% Triton X-100, 1 mM EDTA, 1X leupeptin, 1X aprotinin, and 1X pepstatin) was added. Samples were shaken for 15 min on ice and then collected in 1.5-ml tubes by scraping. After, cells were incubated at 4°C in a rotating wheel for 20 min. Samples were centrifuged at 12,000 $g$ at 4°C for 15 min. For the input, 60 µl of each supernatant was transferred to a new tube, and

12 µl of 6X Laemmli buffer was added. Before the resin was used, 30 µl of Flag-agarose (Cat# A2220; Sigma-Aldrich) resin was dispensed in new 1.5-ml tubes and washed once with 800 µl lysis buffer. The remaining supernatants were added in each tube containing washed flag-agarose resin and incubated in a rotating wheel at 4°C for 1 h. Resins were precipitated by centrifugation at 2,000 *g* for 3 min and washed with 800 µl of lysis buffer four times. Resin pellets were resuspended in 40 µl of 1.2X Laemmli buffer (IP). All the samples were boiled at 95°C for 5 min and spun before western blotting loaded.

### Bioinformatics analysis

The in silico analysis of TANGO2 isoforms and orthologs, structure and acyl interaction, and cavitation predictor was performed using open-source T-Coffee (tcoffee.crg.eu), Alpha-Fold Server (alphafold.ebi.ac.uk), and KVFinder (kvfinder-web.cnpem.br) software, respectively. All the image and movie simulations were created by operating UCSF ChimeraX (cgl.ucsf.edu/chimerax).

### Statistical analysis

Statistical analysis was performed using GraphPad Prism 8.0 (GraphPad) and R software. Data represent the mean ± SDs of N experiments. For multiple comparisons, two-way ANOVA with subsequent Holm–Sidak's posttests was used. P values <0.01 were considered statistically significant.

### Online supplemental material

The supplemental material provides extensive supporting evidence for the study's findings on TANGO2. Fig. S1 illustrates TANGO2 isoform alignments, cellular localization in a different cell type, and protein purification. Table S1 presents detailed cavity analyses of the predictive TANGO2 structure. Five movies offer dynamic visualizations: Video 1 shows TANGO2 isoform 1's mitochondrial localization, Video 2 and Video 3 demonstrate TANGO2's interaction with LDs in wild-type and mutant forms, Video 4 displays TANGO2's 3D structure, Video 5 illustrates TANGO2's cavitation predictions, and Video 6 and Video 7 model TANGO2's interaction with palmitate molecules. These materials collectively support and expand upon the main text's conclusions about TANGO2's structure, localization, and function.

### Data availability

All raw data have been submitted to the open repository Zenodo (https://zenodo.org/records/14864387).

## Acknowledgments

We thank all members of the Malhotra laboratory for fruitful discussions and critical reading of the manuscript. We thank the staff of the CRG Advanced Light Microscopy Unit for technical help, colleagues of the CRG QCB Department for valuable debates, and James Rothman and Eric Soupene for advice with protein–lipid interaction experiments. We acknowledge the support of the Spanish Ministry of Science, the Centro de Excelencia Severo Ochoa, and the CERCA Programme/Generalitat de Catalunya. V. Malhotra is an Institució Catalana de Recerca i Estudis Avançats professor at the Centre for Genomic Regulation. This work reflects only the authors' views, and the EU Community is not liable for any use that may be made of the information contained therein.

These results were funded under PID2022-143128NB-I00 project by MCIN/AEI/10.13039/501100011033/FEDER, UE (to V. Malhotra) and have received funding from the European Research Council (ERC) under the European Union's Horizon 2020 research and innovation programme (grant agreement No 951146; to V. Malhotra) and from the European Union's Horizon Europe under the grant agreement No 101062382 (to A.L. Lujan) and have received funding from the National Center for Global Health and Medicine under the grant agreement Intramural Research Fund 22T001 (to H. Shindou), and received funding from MEXT KAKENHI under the grant agreement JP23738333 (T. Hashidate-Yoshida). Open Access funding provided by the Universitat Pompeu Fabra.

Author contributions: A.L. Lujan: conceptualization, data curation, formal analysis, funding acquisition, investigation, methodology, project administration, resources, validation, visualization, and writing—original draft, review, and editing. O. Foresti: conceptualization, investigation, resources, supervision, and writing—review and editing. J. Wojnacki: data curation and formal analysis. G. Bigliani: investigation and writing—review and editing. N. Brouwers: investigation and methodology. M.J. Pena: investigation. S. Androulaki: resources. T. Hashidate-Yoshida: data curation, investigation, visualization, and writing—review and editing. M. Kalyukina: investigation. S.S. Novoselov: investigation, resources, and supervision. H. Shindou: data curation, funding acquisition, investigation, visualization, and writing—original draft. V. Malhotra: conceptualization, data curation, formal analysis, funding acquisition, project administration, supervision, and writing—original draft, review, and editing.

Disclosures: The authors declare no competing interests exist. Views and opinions expressed are however those of the author(s) only and do not necessarily reflect those of the European Union. Neither the European Union not the granting authority can be held responsible for them.

Submitted: 1 October 2024

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

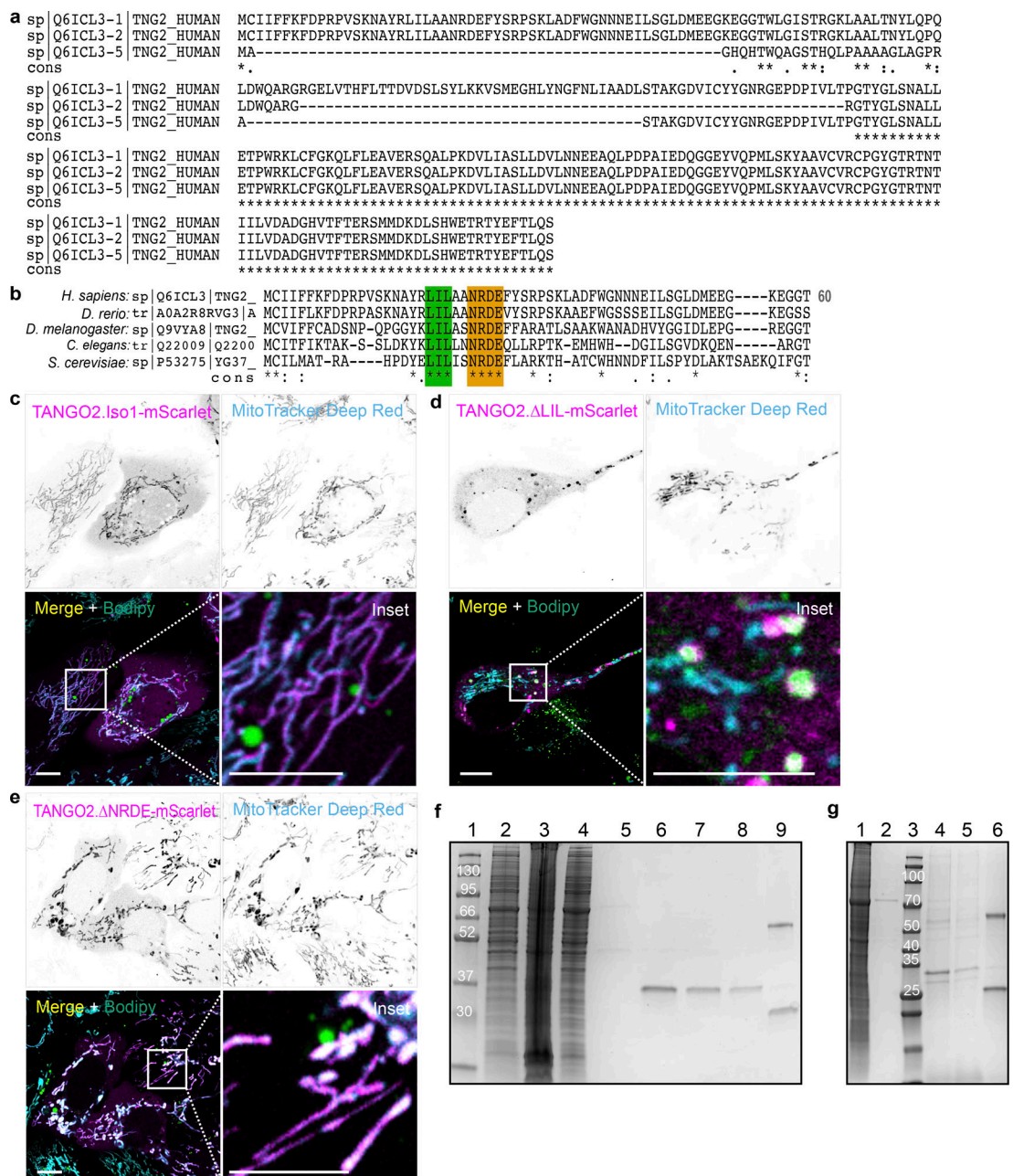

Figure S1. **TANGO2 isoform alignment, location in U2OS cells, and purification in HEK293 cells. (a)** In silico sequence alignments of the full-length TANGO2 isoforms 1 (UniProt ID: Q6ICL3-1), 2 (UniProt ID: Q6ICL3-2), and 5 (UniProt ID: Q6ICL3-5), using T-COFFEE server. Asterisks (*) indicate fully conserved residues, while a colon (:) and a period (.) represent strongly and weakly similar properties in the amino acid sequence. **(b)** Multiple sequence alignment of the 40 amino-terminal residues of TANGO2 orthologs in *Homo sapiens*, *Danio rerio*, *Drosophila melanogaster*, *Caenorhabditis elegans*, and *Saccharomyces cerevisiae*, using T-COFFEE software. The conserved regions LIL (green) and NRDE (orange) are highlighted. **(c–e)**, U2OS cells transfected with TANGO2.Iso1-mScarlet (c), or the mutants TANGO2.ΔLIL-mScarlet (d) and TANGO2.ΔNRDE-mScarlet (e) (magenta) were incubated with the LD marker Bodipy Green and the mitochondrial marker MitoTracker Deep Red (cyan). Squares indicate the magnification area (inset). Scale bars = 10 μm. **(f)** Representative Coomassie staining gel of TANGO2.Flag purification. Lanes were loaded with samples of molecular marker (1, in kDa), cell lysate (2), pellet (3), flow-through fraction after bead binding (4), flow-through fraction after ATP wash (5), first elution (6), second elution (7), third elution (8), and beads (9). **(g)** Representative Coomassie staining gel of TANGO2.ΔNRDE.Flag purification. Lanes were loaded with samples of cell lysate (1), flow-through fraction after ATP wash (2), molecular marker (3, in kDa), first elution (4), second elution (5), and beads (6). Source data are available for this figure: SourceData FS1.

Video 1.   **TANGO2 isoform 1 surrounded by Tom20.** HepG2 cells cotransfected with TANGO2.Iso1-mScarlet (red) and the mitochondrial outer membrane construct Tom20.GFP (green). Cells were incubated with the DNA marker Hoechst-33342 (blue) for 30 min and washed before confocal time-lapse fluorescence microscopy.

Video 2.   **TANGO2 wild-type interaction with LDs.** HepG2 cells cotransfected with the LD marker GPAT4.hairpin-NG (green) and TANGO2.Iso1-mScarlet (red) were incubated for 30 min with Hoechst-33342 (blue) to detect DNA, followed by live-cell confocal imaging.

Video 3.   **TANGO2 mutant interaction with LDs.** HepG2 cells cotransfected with the LD marker GPAT4.hairpin-NG (green) and TANGO2.ΔLIL-mScarlet (red) were incubated for 30 min with Hoechst-33342 (blue) to detect DNA, followed by live-cell confocal imaging.

Video 4.   **TANGO2 structure.** TANGO2 isoform 1 (AF-Q6ICL3-F1) three-dimensional model simulation by AlphaFold. The rainbow-colored chain indicates the beginning of the amino-terminal sequence (blue) to the carboxyl-terminal sequence (red) of TANGO2 visualized using UCSF ChimeraX.

Video 5.   **TANGO2 cavitation and pocket predictions.** Cavitation volume (violet) analysis of the TANGO2 isoform 1 surface (gray) by KVFinder predictor visualized using UCSF ChimeraX.

Video 6.   **TANGO2 structure interacting with one palmitate molecule.** The LIL (green) and NRDE (orange) domain analysis in the TANGO2 isoform 1 surface and its modeling interaction with one palmitate molecule predicted by AlphaFold 3.0 and visualized using UCSF ChimeraX.

Video 7.   **TANGO2 structure interacting with two palmitate molecules.** The LIL (green) and NRDE (orange) domain analysis in the TANGO2 isoform 1 surface and its modeling interaction with two palmitate molecules predicted by AlphaFold 3.0 and visualized using UCSF ChimeraX.

**Provided online is Table S1. Table S1 shows the identification (A to I) and measurement analysis of each cavitation's area, volume, and deepness in angstrom (Å) by KVFinder predictor.**

