## [Peer Review File · The Journal of Cell Biology]

TANGO2 is an acyl-CoA binding protein

Agustin Lujan, Ombretta Foresti, José Wojnacki, Gonzalo Bigliani, Nathalie Brouwers, Maria Pena, Stefania Androulaki, Tomomi Hashidate-Yoshida, Maria Kalyukina, Sergey Novoselov, Hideo Shindou, and Vivek Malhotra

Corresponding Author(s): Vivek Malhotra, CRG, Centre de Regulacio Genomica

Review Timeline:

Submission Date:	2024-10-01
Editorial Decision:	2024-11-12
Revision Received:	2024-12-17
Editorial Decision:	2025-01-21
Revision Received:	2025-01-23

Monitoring Editor: Jodi Nunnari

Scientific Editor: Andrea Marat

Transaction Report:

DOI: <https://doi.org/10.1083/jcb.202410001>

November 12, 2024

Re: JCB manuscript #202410001

Vivek Malhotra
CRG, Centre de Regulacio Genomica

Dear Vivek,

Thank you for submitting your manuscript entitled "TANGO2 transports acyl-CoA into mitochondria for lipid metabolism". Your manuscript has been assessed by two expert reviewers, whose comments are appended below. Unfortunately we did not obtain comments from the third reviewer.

You will see that while the reviewers appreciate the potential impact of your findings, they have significant concerns that in its current form the manuscript is too preliminary for JCB. Please let us know if you are able to address the major issues outlined above and wish to submit a revised manuscript to JCB. Note that a substantial amount of additional experimental data likely would be needed to satisfactorily address the concerns of the reviewers. The typical timeframe for revisions is three to four months. If you anticipate any difficulties in meeting this aforementioned revision time limit, please contact us and we can work with you to find an appropriate time frame for resubmission. Please note that papers are generally considered through only one revision cycle, so any revised manuscript will likely be either accepted or rejected.

However, we understand the time constraints and competition faced, and if you would like to pursue a more rapid publication can offer you to transfer your study to our not-for-profit open-access sister journal, Life Science Alliance (LSA). We shared your manuscript and the accompanying reviews with LSA Executive Editor, Eric Sawey, who is interested in these findings, and would like to publish this manuscript at LSA pending the following revisions:

- Address Reviewer 1's comments #2 & 4.
- Address Reviewer 2 by mentioning the points made as limitations and toning down the related claims.

You may use the link below to transfer your manuscript to LSA. You do not need to revise the manuscript before transferring it to LSA. Once you transfer, Dr. Sawey will email you an invitation to revise and resubmit, listing the same revision requests as mentioned above. Please feel free to reach out at e.sawey@life-science-alliance.org if you have any questions about the LSA journal, the transfer process or the revisions requested.

You also have the option to transfer your manuscript to Molecular Biology of the Cell or Journal of Cell Science. Although we have not discussed your paper with editors at these journals, you will find the option to easily transfer your manuscript files to either journal at the link. Finally, our journal office will transfer the reviews to any external journal upon request.

Link Not Available

Thank you for your interest in Journal of Cell Biology.

Sincerely,

Jodi Nunnari, Ph.D.
Editor-in-Chief

Andrea L. Marat, Ph.D.
Deputy Editor

Journal of Cell Biology

Reviewer #1 (Comments to the Authors (Required)):

The work by Lujan et al. investigates the molecular mechanism by which Tango2 contributes to disease pathology in TDD. The authors describe a new function of tango2 as a carrier of acyl-coA for lipid oxidation in mitochondria. Further experimental support on the mechanism of this transport process will be very helpful in understanding the disease pathology in TDD and improving the understanding of the fatty acid oxidation pathway.

1. Figure 1 shows that the localization of tango2 overlaps with mitochondria; however, many areas in Figure 1A also show distinct localization domains of tango2 that do not overlap with mitochondria. What are these areas? Are these MAMs or some

other subcellular structures?

2. Authors need to explain how studies in HepG2 cell lines explain the phenotypes observed in cardiac and skeletal muscle in TDD as they are differentiated cells vs highly proliferating liver cells. Could there be different disease mechanisms in different cell types?
3. Acyl-coA enters the mitochondria through CAT. What happens to CAT-mediated transport in tango2 deficiency?
4. Authors speculate that tango2 forms a complex with the classic acyl-coA transport machinery, but there is the alternative possibility of CAT-independent acyl coA transport domains that include tango2 within the mitochondria, which may explain the rescue of tango2 partially by an increase in acyl-CoA by vitamin B5. Could they provide evidence for the one way or the other?

Reviewer #2 (Comments to the Authors (Required)):

This study investigates the function of TANGO2, which remains controversial. A recent paper suggested it could be a heme transporter, but this has been questioned. There is some evidence that lipid hemostasis and perhaps ROS signaling is disrupted by mutations in TANGO2. This study shows that some TANGO2 isoforms are inside mitochondria. It presents evidence that TANGO2 binds acyl-CoA. It also suggests acyl-CoA binding by TANGO2 plays a role in mitochondrial lipid metabolism. While this is reasonable, there is no evidence to support this claim or even a hint of how mitochondrial lipid metabolism changes when TANGO2 is mutated. Without some insight, this study is too preliminary for JCB even as a Report. There are two ways the study could be improved.

1. The manuscript suggests "TANGO2 facilitates the trafficking of acyl-CoA to mitochondria." The study would be stronger if rates of import of intact acyl-CoA into mitochondria were assessed (that is, import independent of CPT1). How much import slows after TANGO2 knockdown could then be determined. I appreciate that measuring rates import is challenging.
2. An important control for the lipid binding assay is to show that NBD-acyl-CoA binding can be competed off with acyl-CoAs without NBD. These experiments will also allow estimations of the affinity of TANGO2 for acyl-CoAs with different acyl chains. The ability of other lipids, like acylcarnitine or heme, to compete with NBD-acyl-CoA could also be determined. How certain are authors that acyl-CoA is the most physiologically relevant lipid TANGO2 binds? Once the Kd for various acyl-CoAs is known, it should be possible to determine whether the affinity of TANGO2 for acyl-CoAs is high enough that it could bind the probably low levels of free acyl-CoA in the mitochondrial matrix or inter-membrane space.

Dear Editor and reviewers,

We appreciate your feedback and have made every effort to address your concerns. Your suggestions have significantly improved the quality of our paper, and we believe we have clarified the message regarding the location and lipid-binding properties of TANGO2. The main points we want to emphasize are that TANGO2 is located in the mitochondrial lumen and binds acyl-CoA. To reflect this, we have updated the title of our paper to: TANGO2 is an acyl-CoA binding protein. We have also revised the text extensively to avoid discussions of its mechanism of trafficking to the mitochondrial lumen, and we address this issue regarding its mode of transport to the mitochondria in the last paragraph of the discussion. Our comments, in italics, follow the reviewers' general and specific comments

Reviewer #1 (Comments to the Authors (Required)):

The work by Lujan et al. investigates the molecular mechanism by which Tango2 contributes to disease pathology in TDD. The authors describe a new function of tango2 as a carrier of acyl-coA for lipid oxidation in mitochondria. Further experimental support on the mechanism of this transport process will be very helpful in understanding the disease pathology in TDD and improving the understanding of the fatty acid oxidation pathway.

1. Figure 1 shows that the localization of tango2 overlaps with mitochondria; however, many areas in Figure 1A also show distinct localization domains of tango2 that do not overlap with mitochondria. What are these areas? Are these MAMs or some other subcellular structures?

We have addressed these concerns in our previous paper (Lujan et al. 2023, eLife) by the following independent methods:

1) FRET-based analysis of TANGO2 with Tom20 protein of the mitochondria (Figure 2B-D).

2) We visualized cells expressing TANGO2 tagged with the integral endoplasmic reticulum protein VAP-B and the lipid droplet hairpin protein GPAT4 in all confocal planes (XY, XZ, and YZ). These markers are known to be part of the mitochondria-associated membranes (Figure 2—Figure Supplement 1A-B).

Figure 2B-D:

Figure 2—Figure Supplement 1A:

In the current paper, we have analyzed the Pearson correlation coefficient between TANGO2.Iso1-mScarlet (red channel) and the mitochondrial marker Mitotracker®

Green (green channel), obtaining a Pearson = 0.81. We believe that modifying microscopy channels creates the perception of reduced colocalization. However, colocalization is more apparent in the original channel colors:

Altogether, these data show that TANGO2 is mainly found in the mitochondria. It is also found at some of the sites where mitochondria contact ER and the lipid droplets, respectively.

2. Authors need to explain how studies in HepG2 cell lines explain the phenotypes observed in cardiac and skeletal muscle in TDD as they are differentiated cells vs highly proliferating liver cells. Could there be different disease mechanisms in different cell types?

In this study, we used HepG2 and U2OS cells to demonstrate that TANGO2 localization is not cell-type dependent. Consequently, we have used HepG2 cells for several reasons:

- 1) These cells are derived from the liver, which plays a crucial role in lipid metabolism and the transition from glycolysis to lipolysis, depending on nutrient conditions.*
- 2) HepG2 cells retain many characteristics of hepatocytes, making them suitable for studying metabolic disorders.*
- 3) Their highly proliferative nature facilitates easier manipulation and analysis.*
- 4) Studies involving patient-derived fibroblasts (Lujan et al., 2023), Zebrafish (Kim et al., 2023), and Drosophila (Mehranfar et al., 2024) models exhibit similar metabolic and lipid imbalances phenotypes contributing to the disease state.*

We appreciate the reviewer's concern and now state in the discussion that these findings should also be tested in a variety of cells such as cardiomyocytes and muscle cells to ensure that the location and the function of TANGO2 is not cell type specific.

3. Acyl-CoA enters the mitochondria through CAT. What happens to CAT-mediated transport in tango2 deficiency?

We haven't tested this because it will take us into a different direction of how TANGO2 is trafficked to mitochondria and whether there is a defect in CAT mediated traffic of acyl-CoA in TANGO2 deficiency. We will test this in the future. In this paper we have restricted our efforts to show that TANGO2 is localized to mitochondria predominantly, that it uses LIL sequence to stay in mitochondria, and it binds acyl-CoA by recognizing the acyl moiety through the NRDE domain. This addresses the raging controversy based on the published paper claiming it as a heme binding protein.

4. Authors speculate that tango2 forms a complex with the classic acyl-coA transport machinery, but there is the alternative possibility of CAT-independent acyl coA transport domains that include tango2 within the mitochondria, which may explain the rescue of tango2 partially by an increase in acyl-CoA by vitamin B5. Could they provide evidence for the one way or the other?

How TANGO2 enters mitochondria is indeed the next important question for us and the field. We have shown that LIL sequence in TANGO2 is required for its retention in mitochondria, but whether it uses the CPT1 and CPT2 pathway or a novel pathway for entry remains to be tested. Either way, this is an important objective, which will be addressed in the future. To avoid confusion regarding this issue, we have decided to change the title to better reflect its function as an acyl-CoA binding protein.

As stated in our response to the second reviewer, we have added additional data to show that TANGO2 is indeed an acyl-CoA binding protein and we show its preference to bind the acyl moiety of this modified fatty acid. Once bound to acyl-CoA, how it traffics to the mitochondrial lumen is the next goal of our studies. We have now included the following statements in the discussion to outline the limitations of our studies and the obvious next goals.

In summary, our data strongly suggests that TANGO2 is located within the mitochondrial lumen, and it binds acyl-CoA. This positions TANGO2 as a protein involved in acyl-CoA metabolism. However, it remains unclear whether TANGO2

binds acyl-CoA in the cytoplasm and transports it to the mitochondrial lumen for acetyl-CoA production, which would then enter the β -oxidation pathway and support ATP production through the TCA cycle. Additionally, the mechanism by which TANGO2 is trafficked to the mitochondrial lumen is still unknown—whether it uses the carnitine shuttle or another novel pathway has yet to be determined. We also cannot rule out the possibility that TANGO2 directly binds acyl-CoA within the mitochondrial lumen.

Now that we know TANGO2 is an acyl-CoA binding protein, this finding may help explain why loss of TANGO2 function leads to severe pathologies, such as cardiomyopathy and rhabdomyolysis.

Reviewer #2 (Comments to the Authors (Required)):

This study investigates the function of TANGO2, which remains controversial. A recent paper suggested it could be a heme transporter, but this has been questioned.

There is some evidence that lipid hemostasis and perhaps ROS signaling is disrupted by mutations in TANGO2.

Yes, we showed that in our paper published in elife (Lujan et al., 2023).

This study shows that some TANGO2 isoforms are inside mitochondria. It presents evidence that TANGO2 binds acyl-CoA. It also suggests acyl-CoA binding by TANGO2 plays a role in mitochondrial lipid metabolism. While this is reasonable, there is no evidence to support this claim or even a hint of how mitochondrial lipid metabolism changes when TANGO2 is mutated.

Our data show the location of TANGO2 to mitochondria, that it uses LIL sequence for its retention in the mitochondria, and bind acyl-CoA. We now present data-based on reviewer's suggestion that it binds the acyl moiety of acyl-CoA. We have changed the title and the text to avoid any claims of a direct role of TANGO2-acyl-CoA interaction in mitochondrial physiology.

Without some insight, this study is too preliminary for JCB even as a Report.

This statement is harsh. We believe that showing the role of TANGO2 as an acyl-CoA binding protein is an important step in understanding its function and the eventual connection to physiology.

There are two ways the study could be improved.

1. The manuscript suggests "TANGO2 facilitates the trafficking of acyl-CoA to mitochondria." The study would be stronger if rates of import of intact acyl-CoA into mitochondria were assessed (that is, import independent of CPT1). How much import slows after TANGO2 knockdown could then be determined. I appreciate that measuring rates import is challenging.

As stated by the reviewer these are important experiments, but challenging. This would require in vitro reconstitution and we are nowhere near that stage in our analyses.

2. An important control for the lipid binding assay is to show that NBD-acyl-CoA binding can be competed off with acyl-CoAs without NBD. These experiments will also allow estimations of the affinity of TANGO2 for acyl-CoAs with different acyl chains. The ability of other lipids, like acylcarnitine or heme, to compete with NBD-acyl-CoA could also be determined.

Following the reviewer's suggestion, we measured the binding capacity of NBD-acyl-CoA to TANGO2 and its competition with acyl-CoA (without NBD) and coenzyme A alone. As shown in the new Figure 3 (panels I and J), the interaction between TANGO2 and NBD-acyl-CoA is primarily disrupted by competition with 16:0-CoA. We also observed that coenzyme A alone does not compete in this interaction. These results are now presented and discussed in the manuscript.

How certain are authors that acyl-CoA is the most physiologically relevant lipid TANGO2 binds?

We now clearly show that TANGO2 prefers to bind the acyl-moiety in acyl-CoA. Unfortunately, we could only find commercial preparations of palmitoyl and oleoyl-CoA-NBD. We will need to collaborate with experts to address which of the many forms of acyl-CoA can be selected by TANGO2. But the fact that it binds acyl-CoA is an advance over the complete lack of an understanding of the binding partners of TANGO2.

Once the K_d for various acyl-CoAs is known, it should be possible to determine whether the affinity of TANGO2 for acyl-CoAs is high enough that it could bind the probably low levels of free acyl-CoA in the mitochondrial matrix or inter-membrane space.

This is beyond our expertise and the scope of this paper.

It is also important to note that we have identified significant methodological differences regarding the interaction of TANGO2 with heme molecules. As described in the Materials and Methods section of Han S et al. (2023), an interaction assay was performed over a period of up to 6 hours in a buffer at pH 7.9. In contrast, our approach examines the interaction between TANGO2 and acyl-CoA in a buffer at physiological pH (7.4), and this interaction occurs within a few minutes. Clearly, the data connecting TANGO2 to heme is non-physiological. Furthermore, Sandkuhler SE et al. (2023) [doi: 10.1101/2023.11.29.569072] presented clear evidence using multiple model organisms that contradicts the findings of a study proposing heme transport as the primary function of TANGO2 (Sun F et al., 2022; doi: 10.1038/s41586-022-05347-z). Additionally, the Sun et al. paper has been discussed on PubPeer for containing duplicate data across different panels (<https://pubpeer.com/publications/1EADF72C52BD6BBD4A1FF8330D8E38>).

In all, we thank the reviewer for highlighting the limitations of our claims and we have therefore changed the title and the text to reflect the data on the location of TANGO2 and its binding to acyl-CoA. We have now included the following statements in the discussion to outline the limitations of our studies and the obvious next goals.

In summary, our data strongly suggests that TANGO2 is located within the mitochondrial lumen, and it binds acyl-CoA. This positions TANGO2 as a protein involved in acyl-CoA metabolism. However, it remains unclear whether TANGO2 binds acyl-CoA in the cytoplasm and transports it to the mitochondrial lumen for acetyl-CoA production, which would then enter the β -oxidation pathway and support ATP production through the TCA cycle. Additionally, the mechanism by which TANGO2 is trafficked to the mitochondrial lumen is still unknown—whether it uses the carnitine shuttle or another novel pathway has yet to be determined. We also cannot rule out the possibility that TANGO2 directly binds acyl-CoA within the mitochondrial lumen.

Now that we know TANGO2 is an acyl-CoA binding protein, this finding may help explain why loss of TANGO2 function leads to severe pathologies, such as cardiomyopathy and rhabdomyolysis.

We thank you again for your support and hope the manuscript is now suitable for publication.

Sincerely yours,

Vivek Malhotra

January 21, 2025

RE: JCB Manuscript #202410001R-A

Vivek Malhotra
CRG, Centre de Regulacio Genomica

Dear Vivek,

Thank you for submitting your revised manuscript entitled "TANGO2 is an acyl-CoA binding protein". You will see that the reviewers are overall satisfied with your revisions and responses. Therefore, we would be happy to publish your paper in JCB pending responding to all of Reviewer #2's comments and modifying your discussion as requested, along with final revisions necessary to meet our formatting guidelines (see details below).

A. MANUSCRIPT ORGANIZATION AND FORMATTING:

- 1) Text limits: Character count for Articles is < 40,000, not including spaces. Count includes abstract, introduction, results, discussion, and acknowledgments. Count does not include title page, figure legends, materials and methods, references, tables, or supplemental legends.
- 2) Figures limits: Articles may have up to 10 main text figures.
- 3) * Figure formatting: Scale bars must be present on all microscopy images, including inset magnifications. Molecular weight or nucleic acid size markers must be included on all gel electrophoresis. Aspect ratios of images may not be altered.*
- 4) Statistical analysis: Error bars on graphic representations of numerical data must be clearly described in the figure legend. The number of independent data points (n) represented in a graph must be indicated in the legend. Statistical methods should be explained in full in the materials and methods. For figures presenting pooled data the statistical measure should be defined in the figure legends. Please also be sure to indicate the statistical tests used in each of your experiments (either in the figure legend itself or in a separate methods section) as well as the parameters of the test (for example, if you ran a t-test, please indicate if it was one- or two-sided, etc.). Also, if you used parametric tests, please indicate if the data distribution was tested for normality (and if so, how). If not, you must state something to the effect that "Data distribution was assumed to be normal but this was not formally tested."
- 5) Abstract and title: The abstract should be no longer than 160 words and should communicate the significance of the paper for a general audience. The title should be less than 100 characters including spaces. Make the title concise but accessible to a general readership.
- 6) Materials and methods: Should be comprehensive and not simply reference a previous publication for details on how an experiment was performed. Please provide full descriptions in the text for readers who may not have access to referenced manuscripts.
- 7) All antibodies, cell lines, animals, and tools used in the manuscript should be described in full, including accession numbers for materials available in a public repository such as the Resource Identification Portal. Please be sure to provide the sequences for all of your primers/oligos and RNAi constructs in the materials and methods. You must also indicate in the methods the source, species, and catalog numbers (where appropriate) for all of your antibodies. Please also indicate the acquisition and quantification methods for immunoblotting/western blots.
- 8) Microscope image acquisition: The following information must be provided about the acquisition and processing of images:
 - a. Make and model of microscope
 - b. Type, magnification, and numerical aperture of the objective lenses
 - c. Temperature
 - d. Imaging medium
 - e. Fluorochromes
 - f. Camera make and model
 - g. Acquisition software
 - h. Any software used for image processing subsequent to data acquisition. Please include details and types of operations

involved (e.g., type of deconvolution, 3D reconstitutions, surface or volume rendering, gamma adjustments, etc.).

10) Supplemental materials: There are strict limits on the allowable amount of supplemental data. Articles may have up to 5 supplemental figures. Please also note that tables, like figures, should be provided as individual, editable files. A summary of all supplemental material should appear at the end of the Materials and methods section.

13) ORCID IDs: ORCID IDs are unique identifiers allowing researchers to create a record of their various scholarly contributions in a single place. Please note that ORCID IDs are now *required* for all authors. At resubmission of your final files, please be sure to provide your ORCID ID and those of all co-authors.

Please note that JCB now requires authors to submit Source Data used to generate figures containing gels and Western blots with all revised manuscripts. This Source Data consists of fully uncropped and unprocessed images for each gel/blot displayed in the main and supplemental figures. Since your paper includes cropped gel and/or blot images, please be sure to provide one Source Data file for each figure that contains gels and/or blots along with your revised manuscript files. File names for Source Data figures should be alphanumeric without any spaces or special characters (i.e., SourceDataF#, where F# refers to the associated main figure number or SourceDataFS# for those associated with Supplementary figures). The lanes of the gels/blots should be labeled as they are in the associated figure, the place where cropping was applied should be marked (with a box), and molecular weight/size standards should be labeled wherever possible.

Journal of Cell Biology now requires a data availability statement for all research article submissions. These statements will be published in the article directly above the Acknowledgments. The statement should address all data underlying the research presented in the manuscript. Please visit the JCB instructions for authors for guidelines and examples of statements at (<https://rupress.org/jcb/pages/editorial-policies#data-availability-statement>).

B. FINAL FILES:

****It is JCB policy that if requested, original data images must be made available to the editors. Failure to provide original images upon request will result in unavoidable delays in publication. Please ensure that you have access to all original data images prior to final submission.****

****The license to publish form must be signed before your manuscript can be sent to production. A link to the electronic license to publish form will be sent to the corresponding author only. Please take a moment to check your funder requirements before choosing the appropriate license.****

Thank you for your attention to these final processing requirements. Please revise and format the manuscript and upload materials within 7 days. If you need an extension for whatever reason, please let us know and we can work with you to determine a suitable revision period.

Thank you for this interesting contribution, we look forward to publishing your paper in Journal of Cell Biology.

Sincerely,

Jodi Nunnari, Ph.D.
Editor-in-Chief

Andrea L. Marat, Ph.D.
Deputy Editor

Journal of Cell Biology

Reviewer #1 (Comments to the Authors (Required)):

The authors have addressed all the concerns raised during the revision. I have no further concerns.

Reviewer #2 (Comments to the Authors (Required)):

This manuscript has been focused and improved. It now provides definitive evidence that TANGO2 is in mitochondria and binds acyl-CoAs. These are significant advances in our understanding of TANGO2 function, though the study would be stronger if it had some insight into how acyl-CoA binding by TANGO2 affects lipid metabolism inside or outside mitochondria. Here are some comments.

1. The competition assay can now be used to determine the binding affinity of TANGO2 for acyl-CoAs by adding them at a range of concentrations. Knowing the Kds might be helpful. I'm surprised the affinity of TANGO2 for many lipids was not investigated, especially heme. I understand that lipid binding assays are often challenging since for detergents could be necessary. For future work, ITC might a better choice to assess lipid binding.
2. The manuscript makes two suggestions about TANGO2 trafficking that seem implausible to me. One is that TANGO2 might use the carnitine shuttle. How would a protein use this shuttle that is only known to move fatty acids? The second is that TANGO2 could transport acyl-CoA into mitochondria. Wouldn't the protein need to be unfolded to move into mitochondria? Also, what is the evidence that TANGO2 cycles in and out of mitochondria, couldn't import be only a one-way trip?
3. I am a bit surprised there is not more discussion of how acyl-CoA binding by TANGO2 might affect lipid metabolism. Since most acyl-CoAs are relatively soluble in the aqueous phase, it seems unlikely that TANGO2 (or other acyl-CoA binding proteins) can significantly speed the rate of acyl-CoA movement through the aqueous phase. Instead, TANGO2 could present acyl-CoAs to enzymes that use them. This could enhance enzyme rate or, more likely, channel acyl-CoAs to specific acyl-CoA consuming pathways. This could be discussed.

Dear Editor and reviewers,

We thank you all for your comments and support. We have revised the discussion as per reviewer #2 suggestions. Our comments, in italics, follow the reviewers' general and specific comments.

Reviewer #1 (Comments to the Authors (Required)):

The authors have addressed all the concerns raised during the revision. I have no further concerns.

Thanks for your support.

Reviewer #2 (Comments to the Authors (Required)):

This manuscript has been focused and improved. It now provides definitive evidence that TANGO2 is in mitochondria and binds acyl-CoAs. These are significant advances in our understanding of TANGO2 function, though the study would be stronger if it had some insight into how acyl-CoA binding by TANGO2 affects lipid metabolism inside or outside mitochondria. Here are some comments.

1. The competition assay can now be used to determine the binding affinity of TANGO2 for acyl-CoAs by adding them at a range of concentrations. Knowing the Kds might be helpful. I'm surprised the affinity of TANGO2 for many lipids was not investigated, especially heme. I understand that lipid binding assays are often challenging since for detergents could be necessary. For future work, ITC might a better choice to assess lipid binding.

Yes, we will continue to use this assay to screen the binding partners more broadly and to determine binding affinities.

2. The manuscript makes two suggestions about TANGO2 trafficking that seem implausible to me. One is that TANGO2 might use the carnitine shuttle. How would a protein use this shuttle that is only known to move fatty acids? The second is that TANGO2 could transport acyl-CoA into mitochondria. Wouldn't the protein need to be unfolded to move into mitochondria? Also, what is the evidence that TANGO2 cycles in and out of mitochondria, couldn't import be only a one-way trip?

We have already shown that deletion of the LIL domain causes TANGO2 to be excluded from the mitochondria. It is also important to note that upon starvation, the cytoplasmic pool of TANGO2 increases in the presence of cyclohexamide. This suggests that it moves between the cytoplasm and the mitochondria. However, we agree that it remains to be seen whether the protein cycles or has a unidirectional pathway, so we have added these thoughts to the discussion.

3. I am a bit surprised there is not more discussion of how acyl-CoA binding by TANGO2 might affect lipid metabolism. Since most acyl-CoAs are relatively soluble in the aqueous phase, it seems unlikely that TANGO2 (or other acyl-CoA binding proteins) can significantly speed the rate of acyl-CoA movement through the aqueous phase. Instead, TANGO2 could present acyl-CoAs to enzymes that use them. This could enhance enzyme rate or, more likely, channel acyl-CoAs to specific acyl-CoA consuming pathways. This could be discussed.

Thus is precisely our thinking that it likely presents acyl-coAs to enzymes for downstream events involved in lipid metabolism. These suggestions are now included in the manuscript as below.

“However, it remains unclear whether TANGO2 binds acyl-CoA in the cytoplasm and transports it to the mitochondrial lumen to support the β -oxidation pathway, facilitates the presentation of acyl-CoA to specific enzymes involved in lipid metabolism, traffic to the mitochondrial lumen via the carnitine shuttle pathway, or utilizes a novel, stress-induced pathway.”

We thank you again for your support and hope the manuscript is now suitable for publication.

Sincerely yours,

Vivek Malhotra